# A four-year-old can outperform ResNet-50: Out-of-distribution robustness may not require large-scale experience

Lukas S. Huber[1-2]§        Robert Geirhos[2-3]        Felix A. Wichmann[2]

[1]University of Bern
[2]University of Tübingen
[3]International Max Planck Research School for Intelligent Systems
§To whom correspondence should be addressed: `lukas.huber@unibe.ch`

## Abstract

Recent gains in model robustness towards out-of-distribution images are predominantly achieved through ever-increasing large-scale datasets. While this approach is very effective in achieving human-level distortion robustness, it raises the question of whether human robustness, too, requires massive amounts of experience. We therefore investigated the developmental trajectory of human object recognition robustness by comparing children aged 4–6, 7–9, 10–12, 13–15 against adults and against different deep learning models. Assessing how recognition accuracy degrades when images are distorted by salt-and-pepper noise, we find that while overall performance improves with age, even the youngest children in the study showed remarkable robustness and outperformed standard CNNs and self-supervised models on distorted images.

In order to compare the robustness of different age groups and models as a function of visual experience, we used a back-of-the-envelope calculation to estimated the number of 'images' that those young children had been exposed to during their lifetime. We find that for humans, more data does not necessarily lead to better out-of-distribution robustness. Compared to various deep learning models, children's high out-of-distribution robustness requires relatively little data. Taken together, this indicates that human out-of-distribution robustness develops very early in life and may not require seeing billions of different images during lifetime given the right choice of representation and information processing optimised during evolution.

## 1   Introduction

Not too long ago, humans outperformed convolutional neural networks (CNNs) on out-of-distribution (OOD) datasets [e.g., see 1–5]. However, recently this longstanding robustness gap between humans and CNNs is beginning to close. Currently, the best state-of-the-art (SOTA) models are matching, or even exceeding human performance on most OOD datasets [6]. Even though some of these models have an innovative architecture and/or training procedure—such as CLIP [7] or other implementations of vision transformers [8]—the most crucial feature to achieve human-like OOD robustness appears to be *large-scale training*. While standard training on ImageNet includes 1.3M images, most models showing human-like OOD robustness are trained on much larger datasets—ranging from 14M (Big Transfer models [9]) to staggering 940M images (semi-weakly supervised models [10]). While

3rd Workshop on Shared Visual Representations in Human and Machine Intelligence (SVRHM 2021) of the Neural Information Processing Systems (NeurIPS) conference, Virtual.

both architecture and data matter—vision transformers, e.g., trained on ImageNet are better than standard CNNs trained on ImageNet—even standard CNNs trained on large-scale datasets (such as Big Transfer models) achieve remarkable robustness. This indicates that large-scale training may be *sufficient* for OOD robustness. Here we ask whether large-scale training is also *necessary* to achieve OOD robustness. If so, we would expect human OOD robustness to be low in early childhood, and to increase with age (= more input) (hypothesis I). Alternatively, human OOD robustness might instead result from clever information processing and representation that copes with less data. In this case, we would expect human OOD robustness to be already high in early childhood (hypothesis II).

Both hypotheses can be evaluated with respect to developmental data. However, behavioral research investigating the development of object recognition (robustness) in children (after 2 years of age) and adolescents is extremely sparse and several reviews have pointed out the lack of such studies [11–13]. Here, we present a detailed investigation of the developmental trajectory of object recognition robustness. We gathered 7,200 psychophysical trials from children, adolescents and adults and compared their performance with that of various deep learning models. We then looked at children's OOD robustness as a function of the estimated number of input images (i.e., dataset-size) at different time points during development. In doing so we thus investigated whether human OOD robustness develops early during development and if—similarly to some SOTA models—human OOD robustness is dependent on large-scale visual input.

## 2   Methods

**Psychophysical experiment.** The methods used in this study are adapted from a series of psychophysical experiments conducted in the Wichmann-lab [2]. Some aspects were slightly modified in order to get a paradigm suitable to test children (see Appendix A.1 for more information). To obtain an OOD dataset, we added salt-and-pepper noise to 320 ImageNet-images (see Appendix A.2 for more details on the dataset). Figure 1 shows some example stimuli. In total, we collected 7,200 trials from a sample of 46 children/adolescents (4–15 years) and three adults. The study was approved by the institutional ethical review board of the University of Bern. All participants reported normal or corrected to normal vision and provided (parental) written consent. In each trial, participants were presented with an image and had to choose the corresponding category out of 16 basic entry-level categories (such as cat, knife, truck, etc.). Stimuli were balanced with regard to those categories and presented in a pseudo-random order (see Appendix A.2 for more information). To compare humans and models we mapped the 1,000 class decision vector of ImageNet-trained models to the 16 classes using the WordNet hierarchy [14]. Stimuli ($256 \times 256$ pixels) were presented at the centre of a 14" screen (spatial resolution of $1920 \times 1200$ pixels at a refresh rate of 120 Hz) for 300 ms. Given a viewing distance of 60 cm, retinal image size was $4 \times 4$ degrees of visual angle. Stimuli were then followed by a pink noise mask with $1/f$ spectral shape. Short presentation time and noise mask was employed to ensure fair comparison between humans and feedforward neural networks.

**Models.** Using the model-vs-human toolbox [6], we evaluated different models using the same stimuli as we used to test human observers: The evaluated models were chosen to be representative for different classes of deep learning models: AlexNet [15] and ResNet-50 [16] as standard CNNs (both trained on ImageNet); a Big Transfer model [9], Noisy Student [17] and a semi-weakly supervised model [10] as representatives for standard models trained on substantially larger datasets; SimCLR [18] as a current self-supervised model, a vision transformer [8] as a current model with a more powerful architecture and CLIP [7] as an instance of a model trained on a large-scale dataset and a non-standard architecture. For further details on methods and models refer to Appendix A & C.

## 3   Result I (psychophysical experiment): Developmental trajectory—human OOD robustness develops early

In order to assess the development of human OOD robustness we measured classification accuracy as a function of degradation level for all age groups. To compare the relative decrease in accuracy between the different observers we further normalised classification accuracy with respect to the initial accuracy at degradation level zero. The results are shown in Figure 2. To put these trajectories into perspective we also include two models—one SOTA model trained on a large-scale dataset (CLIP) and a standard CNN trained on ImageNet (ResNet-50) in the following plots. A detailed comparison between models and human observers is given in Section 4.

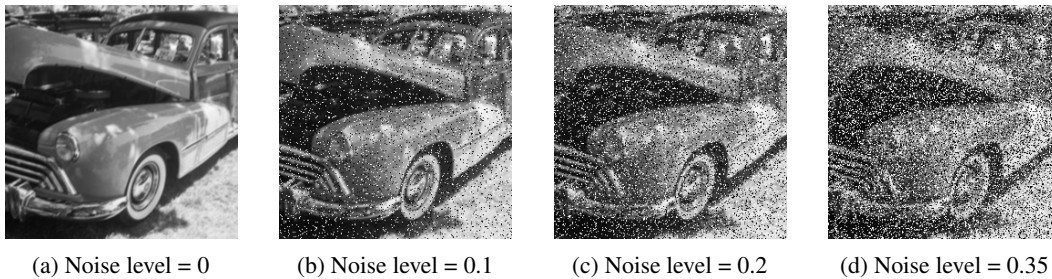

| (a) Noise level = 0 | (b) Noise level = 0.1 | (c) Noise level = 0.2 | (d) Noise level = 0.35 |

Figure 1: Systematic degradation of images with salt-and-pepper noise. We employed 4 different proportions of flipped pixels (0, 0.1, 0.2 or 0.35). For example, 0.2 means that 20% of the pixels are switched while 80% of the pixels remain untouched. Note that even though different degradation levels are shown for the same image, participants never encountered the same image more than once to avoid memory confounds.

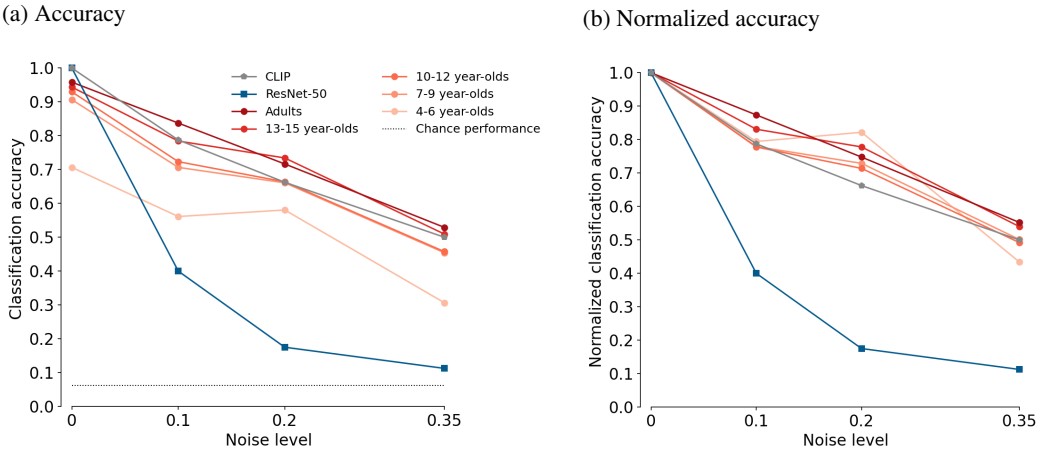

Figure 2: Top 1 classification accuracy (a) and normalised classification accuracy (b) for different age groups, as well as for CLIP and ResNet-50. Both models were evaluated on the same images as human observers. The dotted line at the bottom represents chance level performance of 6.25% (100% divided by the number of categories, which was 16). Normalised accuracy shows the change in accuracy relative to the initial obsever group specific accuracy at difficulty level zero.

**Overall accuracy.** Results indicate that object recognition of 4–6 year-olds is already remarkable robust against parametric image degradations. Accuracy decreased as a function of noise level (as expected) and increased as a function of age. Increase was largest between the 4–6 and 7–9 year-olds. The latter group and the remaining age groups showed accuracies comparable to the one of adults. However, and most importantly, already 4–6 year-olds outperformed ResNet-50 on very slightly distorted images, even in absolute performance. Object recognition robustness of young children thus seems to be much more similar to that of adults than that of ResNet-50. This becomes even clearer when we look at normalised accuracy. Here, we see that relative to their initial accuracy, even 4–6 year-olds demonstrate an adult-like robustness trajectory. In other words, while the younger children are not yet as good at recognising objects as the older age groups (as reflected by gap in absolute classification accuracy), their relative robustness seems already adult-like (as reflected by similar normalised accuracy across difficulty levels). CLIP shows—in contrast to ResNet-50—high OOD robustness and a human-like robustness trajectory w.r.t. increasing image degradations. Thus, Figure 2 illustrates that human OOD robustness develops very early and is already in place by the age of four.

**Confusion matrices.** Moving from the overall classification accuracy (coarse-grained) to classification accuracy and error patterns on the category-level (more fine-grained), we find a similar pattern

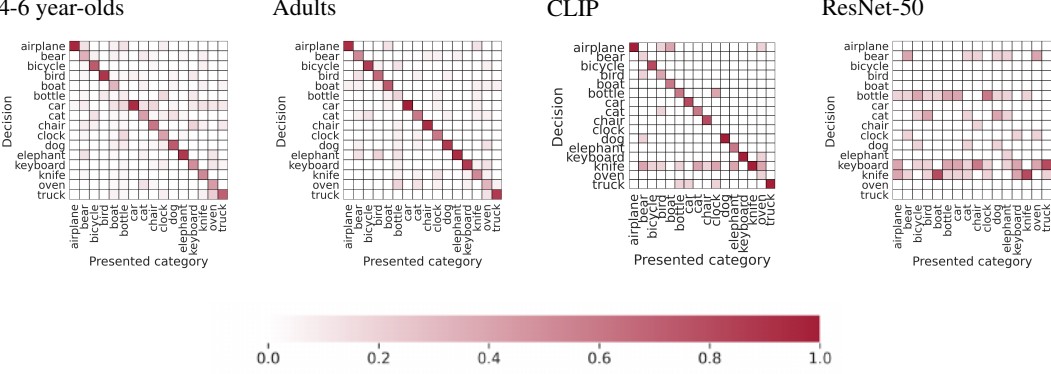

Figure 3: Confusion matrices for 4–6 year-olds, adults, CLIP and ResNet-50 in the salt-and-pepper noise condition at a signal-to-noise ratio of 4:1 (noise level = 0.2). Rows show classification decision of observers and CNNs and columns show the ground truth label of the presented category. Transparency of single squares within a matrix represent response probabilities (see color bar at the bottom). Entries along the negative diagonal represent correct responses; entries which are off the negative diagonal indicate errors.

of results. Figure 3 shows confusion matrices of children, adults and both models for moderately distorted images. While ResNet-50 is biased towards a small subset of categories and shows a tendency to categorise images regardless of the ground truth category (similar to [2, 19]), this is not the case for human observers. Just like adults, young children show high accuracies along the negative diagonal and do not favour a small subset of categories when making errors. Additionally, young children and adults find similar categories easy to recognise (e.g., airplane, bicycle, car, elephant, etc.). These similarities further indicate that adult-like object recognition robustness emerges early in life. Again CLIP is more similar to human observers (irrespective of age) than to a standard trained CNN. The full set of confusion matrices (all age groups and all noise levels) can be found in Appendix D.

## 4  Results II (comparison with different models): A four-year-old can outperform several deep learning models—more data does not necessarily lead to better OOD robustness

So far, we have seen that object recognition emerges early and is largely in place by the age of four—but in order to put these results into context and relating them to models trained on large-scale datasets, we will need to estimate the number of "images" that young children are exposed to during their lifetime and compare different age groups with different models.

**Basic idea.** First, we performed a back-of-the-envelope calculation to attain the number of input images children are exposed to at different points during development. Next, we evaluated several models which differ w.r.t. training data and/or architecture on the salt-and-pepper OOD dataset. We then compared models and different age groups regarding OOD robustness.

**Estimating number of input images.** We estimated the number of input images by calculating the total number of fixations during lifetime for each age group. In order to do so, we made two assumptions: (a) accumulated wake time for any given age group and (b) fixations per second for any given age group. Calculating the former is straightforward: During development, wake time gradually increases as a function of age. For example 0–1 year-olds are, on average, awake for 11.5 hours a day, whereas adults are awake for 16.5 hours [20]. We took the mean age of each tested age group and calculated the total accumulated wake time for this particular age in seconds. However, estimating the number of fixations per second is more difficult: Fixation duration varies to a great extent [100–2000 ms; e.g., see 21, 22] and is heavily dependent on age and the given visual task [23]. Thus, as a reference, we chose a task which is close to an everyday natural setting (a picture inspection task) and for which developmental data is available [23]. We then calculated fixations per second for each age group based on the fixation duration measured for the mean age of this particular

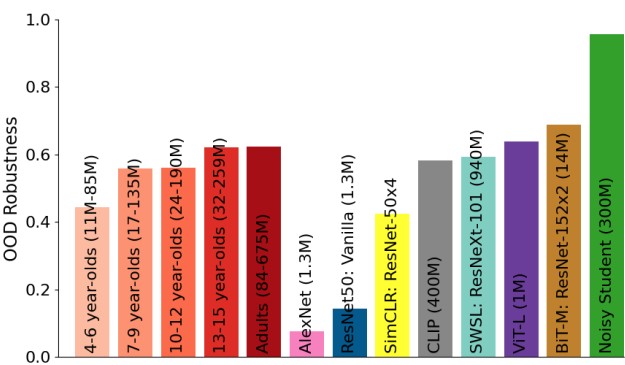

Figure 4: Average OOD robustness and number of input images for different age groups and different models. OOD robustness is calculated as the mean of classification accuracy on moderately and heavily distorted images. Number of input images is given in millions. For human observers these numbers refer to the lower and upper bound of input images as described above.

age group. Because there is no available data for adults in the picture inspection task, we estimated the fixation duration of adults by fitting a linear regression line. Fixations per second calculated in this way ranged from 2.56 for 4–6 year-olds to 3.42 for adults (see Apendix B for details).

However, one may not want to count each fixation as a new input image. Using head-mounted cameras it has been shown that frequency distributions of objects in toddlers' input data are extremely right skewed [24]: Toddlers have only experience with very few objects of a certain category but see those objects (images) very often. It is not clear whether this is just a non-optimal consequence of the natural learning environment of humans or whether the existence of many similar views of the same object plays an important role in object name learning and thus in learning robust visual representations [e.g., see 25]. The later is quite possible given the fact that common data augmentations (such as random crop with flip and resize, colour distortion, and Gaussian blur) lead to higher OOD robustness [e.g., see 26–28]. Furthermore, and similar to human observers, it is not clear how to count input images for deep learning models. Should every augmented image be counted as a new image? It is unlikely that training on a smaller datasets for an increased number of epochs yields the same increase in robustness as training on a larger datasets. Therefore, we used training dataset size as an estimate of the number of input images for deep learning models. Consequently, for fair comparison it may not be correct to estimate the number of input images of human observers by counting *all* fixations. Thus, we scaled down the total number of fixation (*wake time in seconds × fixations per second*) by different factors to obtain a lower (factor: 24—approximately a new image every 8 seconds) and an upper bound (factor: 3—approximately a new image every second) of accumulated input images over lifetime.[1]

**Estimating robustness.** As a global measurement of classification robustness we calculated the mean classification accuracy over all moderately and heavily distorted images (noise level 0.2 and 0.35) for each age group and all models. Figure 4 shows the comparison between different age groups and models regarding OOD robustness and number of input images.

**Interpretation.** The comparison in Figure 4 implies that: First, as pointed out above, humans achieve remarkable OOD robustness already at an early stage during development (4–6 years of age). In order to reach this robustness humans seem to require relatively little visual input. For example, comparing 7–9 year-olds with the semi-weakly supervised model (SWSL: ResNeXt-101) trained on 940M images we can see that humans need 6–24 (depending on whether the lower or upper bound for input images is considered) times less visual input in order to achieve similarly high OOD robustness. This indicates, that while large-scale datasets may be sufficient, they seem not to be necessary to achieve

---

[1]Of course these factors are somewhat arbitrary. However, we feel confident that assuming a completely new image every one to eight seconds is a rather conservative estimate; if anything the lower bound may well be much lower still.

high OOD robustness. A point which is further illustrated by the high OOD robustness of the vision transformer model (ViT-L), which uses little input data but another computational approach. In other words, the human brain, and at least some models, do not solely rely on a large-scale visual input to achieve high OOD robustness but instead may employ a smarter choice of information representation and processing.

Second, human OOD robustness does not increase linearly or exponentially as a function of visual experience; rather there is a logarithmic increase which reaches a ceiling at a certain developmental stage (13–15 years of age). This suggest that, at least for humans, more data does not equal better OOD robustness. Notably, the human robustness ceiling seems not to be a general constraint, as Noisy Student demonstrates that it is possible for a system to go beyond this ceiling. With regard to the human visual system this suggests that there was either no need for better OOD robustness during evolution, or that there are some biological and/or environmental constraints which inhibit the development of higher OOD robustness (such as the statistical properties of the learning environment described above).

Third, 4–6 year-olds not only outperform standard CNNs trained on ImageNet, but also current self-supervised models which are learning by deep contrastive embedding methods [e.g 29]. Zhuang et al. [30] recently employed this approach and showed that self-supervised models trained solely with noisy child developmental data (collected from head-mounted cameras) learn strong visual representations that achieve good neural predictivity on different areas across the ventral visual stream and show reasonable performance on image classification tasks, arguing that self-supervised learning models might be biologically plausible computational models for the visual system. However, others find that in order to match human performance self-supervised models would need million years of natural visual experience [31]. Similarly, in accordance with [19] the present findings indicate that the robustness of human visual representation is not yet met by self-supervised models.

## 5  Discussion

**Summary.** We investigated the emergence of human OOD robustness during development. In order to do so we employed a well-established psychophysical image-category identification task. Until now, investigations of the developmental trajectory of object recognition beyond two years of age have been scarce [11–13] and, to our knowledge, this is the first study to directly compare children, adolescents, and different neural networks in a psychophysical core object recognition task.[2] We then estimated the number of 'images' that young children have been exposed to during their lifetime and compared human observers with different deep learning models. This showed that while large-scale training is very effective in increasing model OOD robustness, large-scale experience may not be required given that human robustness develops very early and after reaching a certain stage does not continue to improve as visual input is accumulated over time.

**Limitations.** The employed experimental set-up did not allow to test children younger than four years of age. It would be interesting to push this further and investigate the developmental trajectory during infancy to further disentangle the influence of evolution vs. lifetime experience. Furthermore, our comparisons did not allow to separate the influence of object view diversity and dataset size. As recent studies have shown that more diverse images of an object result in models that learn more robust object representations [e.g., see 33], it will be interesting to evaluate and compare humans and models regarding OOD robustness for datasets based on real biological datastreams.

**Conclusion and outlook.** The results presented here suggest that the emergence of human OOD robustness takes place very early in development and does not seem to be dependent on large-scale visual input—after a certain point in development, more data does not equal better OOD robustness. That is, large-scale training might be sufficient, but not necessary for OOD robustness. It is more likely that superior information representation and processing—and, perhaps, multi-modal input from vision and touch, or active explorative behaviour of infants—are further crucial ingredients to human OOD robustness. Provided with relatively little data, even 4–6 year-olds outperform standard trained CNNs and self-supervised models on distorted images. Even though the current trend to train models

---

[2]There is one unpublished investigation comparing children and CNNs that was presented at the 20th Annual Meeting of the Vision Sciences Society [32]. The findings presented there are in line with the results of the current study. However, the current study extends these findings by covering a larger age range, using naturalistic images with parameterized distortions and more response categories.

with ever-larger datasets may close the longstanding robustness gap between humans and neural networks [6], it does not seem to capture the learning process by which human object recognition becomes robust. As a proof of concept, humans demonstrate that it is possible to learn much from little data—thus, much may be gained from smarter models, not just larger datasets.

**Acknowledgements**

We would like to thank all other members of the Wichmann-lab for their support and insightful discussions. Special thanks goes to Uli Wannek for superior technical advice and Silke Gramer for extremely kind and patient administrative support. Additionally, we would like to express our gratitude to Gert Westermann and Hannes Rakoczy for their advice and help while designing the children's study and to all children and teachers that participated in the study.

The authors thank the International Max Planck Research School for Intelligent Systems (IMPRS-IS) for supporting Robert Geirhos. Felix Wichmann is a member of the Machine Learning Cluster of Excellence, funded by the Deutsche Forschungsgemeinschaft (DFG, German Research Foundation) under Germany's Excellence Strategy – EXC number 2064/1 – Project number 390727645.

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

# Appendix

We here report experimental details regarding the conducted psychophysical experiment (A), describe details regarding models and evaluation (C), give details on how we estimated the number of input images seen during lifetime (B) and provide the full set of confusion matrices for all age groups and noise levels (D).

# A   Experimental details regarding psychophysical experiments

## A.1   Changes to the initial experimental paradigm

The methods used in this study are adapted from a series of psychophysical experiments conducted in the Wichmann-lab [2]. Here, we largely preserved this experimental procedure and relied on similar stimuli and setup. However, we adapted the following aspects to get a paradigm suitable to test children: We introduced a certain degree of gamification (see Section A.4), added more breaks, and suspended the fixed length of the experiment in order to increase data quality. Children who were motivated were allowed to continue the task, whereas children who were exhausted more quickly were given the possibility to terminate the task after each block of stimuli presentation (20 trials). Additionally, we slightly increased the stimulus presentation duration (from 200 ms to 300 ms) and only used stimuli which were correctly recognised by at least two adults in the previous studies. Finally, children were not tested in our visual psychophysics laboratory but in a quiet place in their school.

## A.2   Dataset

As stimuli we used a subset of images (256×256 pixels) from the 16-class-ImageNet [2], which is a subset of ImageNet [34, license & download at `https://image-net.org/download.php` ]. The 16-class-ImageNet database groups a subset of fine-grained ImageNet categories into 16 entry-level categories using the WordNet hierarchy [14]. In doing so it allows for easy comparison between humans and neural networks. A subset of 521 stimuli—stimuli that were correctly classified by at least two adults in prior experiments—served as a starting point for the present study. We chose this subset because we expected children's performance and motivation to be weaker compared to adults. Therefore, we felt that choosing a subset at least not containing exceedingly ambiguous or even "impossible" images would prevent children from getting frustrated too quickly. We then randomly sampled 320 images (20 for each of the 16 categories) to be manipulated in the next step. A certain proportion of pixels were either set to a gray value of 1 or 0. This manipulation is often referred to as *salt and pepper noise*. We manipulated the images to four different degrees, resulting in 4 different difficulty levels, which corresponded to 4 different proportions of flipped pixels (0, 0.1, 0.2 or 0.35). For example, 0.2 means that 20% of the pixels are switched while 80% of the pixels

remain untouched. The higher the proportion of switched pixels, the more difficult it is to recognise the content (see Figure 1 for an example).

## A.3 Instructions and procedure

Each trial consisted of several phases. First, we presented an attention grabber inspired by an expiring clock (solid white circle which empties itself within 600 ms) in the center of the screen. We chose a moving stimulus instead of a more commonly used fixation cross in order to compensate for possible weaker attention in children. Second, the target image was shown in the center of the screen for 300 ms, followed by a full-contrast pink noise mask ($1/f$ spectral shape) of the same size and duration. Next, the screen turned blank and participants were required to indicate their answers. They did this by physically pointing to one of 16 icons corresponding with the 16 entry-level categories on a laminated DIN A4 sheet arranged in a $4 \times 4$ grid (icon size: $3 \times 3$ cm). We chose this physical response surface mainly for time efficacy (having 4-year-olds handle a computer mouse by themselves can be a lengthy undertaking). Next, the 16 icons appeared on screen, and the experimenter recorded the response using a wireless computer mouse. As in the experiments conducted by Geirhos et al., our icons were a modified version of the ones from the MS COCO website (`https://cocodataset.org/#explore`).

All participants were tested in a quiet room in a single setting—either in their school (children, adolescents) or at home (adults). The experimental session started with some example images and 10 practice trials. For each category, we showed a prototypical example image in the center of the screen and asked participants to name the depicted object. The correct category was indicated by the subsequent presentation of the corresponding category icon. After completing all 16 examples, participants completed 10 practice trials that were identical to the experimental trials, except that undistorted color images were presented. If participants failed on two or more images, they had to do another round of 10 practice trials. Participants were instructed to respond as fast and accurately as possible, and to go with their best guess if unsure. Before the experimental trials started, a single distorted image (matched for the given condition) was shown and a short story-like explanation was given to justify why some of the subsequent images would be distorted ("Somebody spilled salt and pepper; that is why some of them look a bit strange."). Experimental trials were arranged in blocks containing 20 trials each. After each block participants received feedback and were asked whether they would like to continue, have a break or terminate the session. Adults were not explicitly asked if they want to terminate the session—but of course *all* participants were informed at the outset that they could abort the experiment at any given time. Participants could complete a maximum of 16 blocks (320 images). Code for the experiment can be found here.

Table 1: Descriptive statistics of participants and observations split by conditions. Sample size and quantity of observations (*n*), as well as mean (*M*) and standard deviation (*SD*) for age and trials within observer groups. Gender distribution (♂/♀) is given in percentages. Note that for adults, trial *M* equals the total number of trials in the respective condition and trial *SD* is zero because they had to go through all trials of that particular condition.

| Age group | n | Age within group | | ♂/♀ | Trials | | |
| | | M | SD | | n | M | SD |
|---|---|---|---|---|---|---|---|
| 4–6 year-olds | 15 | 5.13 | 0.64 | 33/67 | 1240 | 62.66 | 54.96 |
| 7–9 year-olds | 9 | 8.11 | 0.78 | 47/53 | 1840 | 204.44 | 112.60 |
| 10–12 years-olds | 15 | 11.00 | 0.85 | 47/53 | 1880 | 125.33 | 71.90 |
| 13–15 years-olds | 9 | 14.22 | 0.97 | 44/56 | 1280 | 142.22 | 82.12 |
| Adults | 3 | 28.33 | 5.51 | 33/67 | 960 | 320.00 | 0.00 |

## A.4 Gamification

In order to increase motivation and make the experiment more appealing to children, we gamified several aspects of the experiment. In the beginning, participants could choose one of four characters (matched for gender) corresponding to four different roles: spy, detective, scientist or safari guide. All story lines had the same plot. The chosen character had to undergo a training session designed to

improve his/her crucial skill. What the participants did not know was that the crucial skill, identifying objects as quickly and accurately as possible, was the same for all characters. After each trial, the chosen character was displayed at the foremost position of a progress bar indicating how far the participant had progressed in the current block (level). After each block, participants were provided with feedback designed to be perceptually similar to the display of a game score in an arcade game. There were three different types of scores. Participants received 10 `coins` as a reward for a finished block (not performance related). Additionally, for every two correctly recognised images, they received a `star` (performance-related). If they scored more than eight stars, they earned a special `emblem` matched for the chosen story character.[3] Figure 5 visualises different gamified elements.

### A.5 Descriptive information and participant compensation

The total sample consisted of 46 children and three adults (60% female and 40% male). One child decided to cancel the study right after completing practice trials; therefore, no data was collected for this participant. Descriptive information about the sample and observations is presented in Table 2. We recruited children from 17 different schools and the adult sample was recruited through private contacts. As an incentive for their participation, participants received a book of their choice. The total amount spent on participant compensation amounts to 433.3 Euro.

## B    Estimating number of input images

Table 2: Details regarding the estimation of the number of input images in Section 4. The estimate of accumulated *Wake time* (in millions of seconds) is based on [20]. *Fixation Duration* (in milliseconds) refers to the fixation duration in a picture inspection task [23] and is used to calculate *Fixations per second*. Because there was no available data regarding the fixation duration of adults in this task we assumed a linear relationship between age and fixation duration and used the children's data to fit a simple regression model to estimate the fixation duration of adults ($\hat{y} = -4.33X + 413.66$). Plugging in the mean age of adults (28) yields a predicted fixation duration of 292.42 for adults. *Lower* and *Upper* (in millions of input images) refers to the lower and upper bound of input images encountered during lifetime. The lower bound is calculated by scaling the total number of fixations by 24 and the upper bound by 3. E.g., at the age of five a child has been awake for approximately 99.41 million seconds; it has made about 254.5 million fixations during this time ($99.41 \times 2.56$). Based on these numbers we estimate that a five year old child has most likely not seen less than 10.6 and not more than 84.83 million images (total number of fixations during lifetime either scaled down by a factor of 24 or 3).

| Age group | *M* age | Wake time | Fixation duration | Fixations per second | Lower | Upper |
|---|---|---|---|---|---|---|
| 4–6 year-olds | 5.13 | 99.41 | 390.00 | 2.56 | 10.60 | 84.83 |
| 7–9 year-olds | 8.11 | 154.16 | 380.00 | 2.63 | 16.90 | 135.15 |
| 10–12 year-olds | 11.00 | 211.54 | 370.00 | 2.70 | 23.80 | 190.39 |
| 13–15 year-olds | 14.22 | 272.42 | 350.00 | 2.86 | 32.43 | 259.43 |
| Adults | 28.33 | 591.71 | 292.42 | 3.42 | 84.32 | 674.55 |

## C    Model details

All models were obtained through the `modelvshuman` Python-toolbox [6], where models are referred to as: 'alexnet' (AlexNet), 'resnet50' (ResNet-50: Vanilla), 'simclr_resnet50x4' (SimCLR: ResNet-50x4), 'clip' (CLIP), 'ResNeXt101_32x16d_swsl' (SWSL: ResNeXt-101), 'vit_large_patch16_224' (ViT-L), 'BiTM_resnetv2_152x2' (BiT-M: ResNet-152x2), and 'efficientnet_l2_noisy_student_475' (Noisy Student). The SWSL as well as the SimCLR model use a ResNet-50 architecture and CLIP uses a VitB/32 backbone. Evaluation was done using the following settings: Batch size = 16; number of workers = 4. Because images were grayscale, all three RGB channels were set to be equal to the grayscale image's single channel.

---

[3]The emblems were sunglasses for the spy, a magnifying glass for the detective, a microscope for the scientist, and a camera for the safari guide.

# D Confusion matrices

Figure 6 shows confusion matrices for all observers across all noise levels. Up to and including noise level 0.2, all human observers show very similar confusion patterns. However, at noise level 0.35 the performance of 4–6 year-olds drops significantly. Nevertheless, response probabilities along the negative diagonal did not drop uniformly but remain high for some age-appropriate categories, such as such as `bear`, `bicycle`, `car`, or `elephant`. This indicates that young children´s weaker overall performance is not due to to a generally weaker ability to recognise objects but rather to weak performance on a subset of categories for which robust representations have not yet been acquired.

In comparison to human observers, ResNet-50 becomes biased towards a small subset of categories. That is, there is an emerging tendency to categorize images regardless of the ground truth category. At noise level 0.35 this biased behavior becomes even more pronounced. Almost all images are now categorised either as "bottle" or as "knife" by ResNet-50. CLIP on the other hand shows better overall performance—indicated by the higher response probabilities along the negative diagonal—as well as more human-like response patterns. There is even a similar subset of categories (e.g., `airplane`, `car`, or `keyboard`), which seems easy to recognise for both, CLIP and human observers. However, at noise level 0.35 CLIP starts to become biased towards a certain category (`knife`) as well. In other words, while CLIP shows—compared to ResNet-50—very human-specific behavior, it also seems to have a tendency which is typically associated with standard-trained CNNs. It would be interesting to check whether this tendency becomes more pronounced when the noise level is further increased.

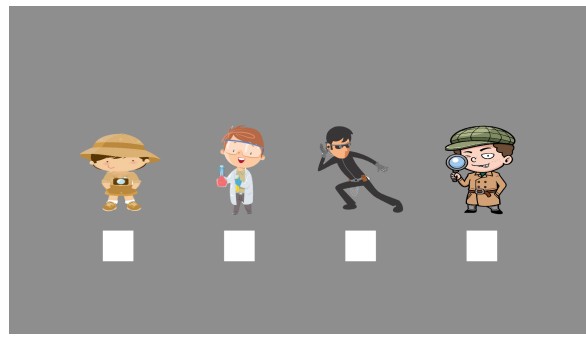

(a) Character selection at the beginning of each session.

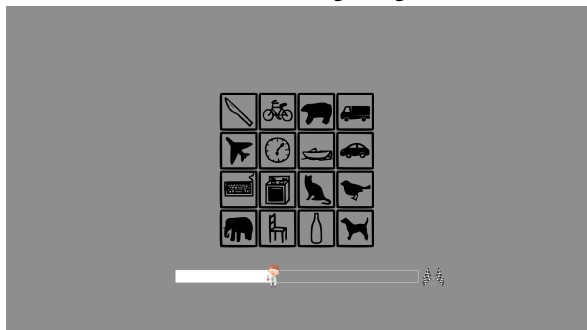

(b) Response screen with gamified progress-bar.

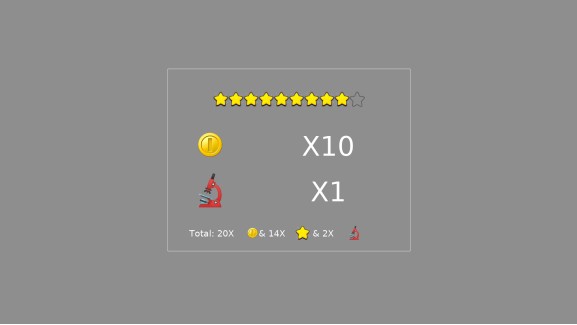

(c) Score display with star- and coin-scores and emblem.

Figure 5: Screenshots at different time-points during the experiment.

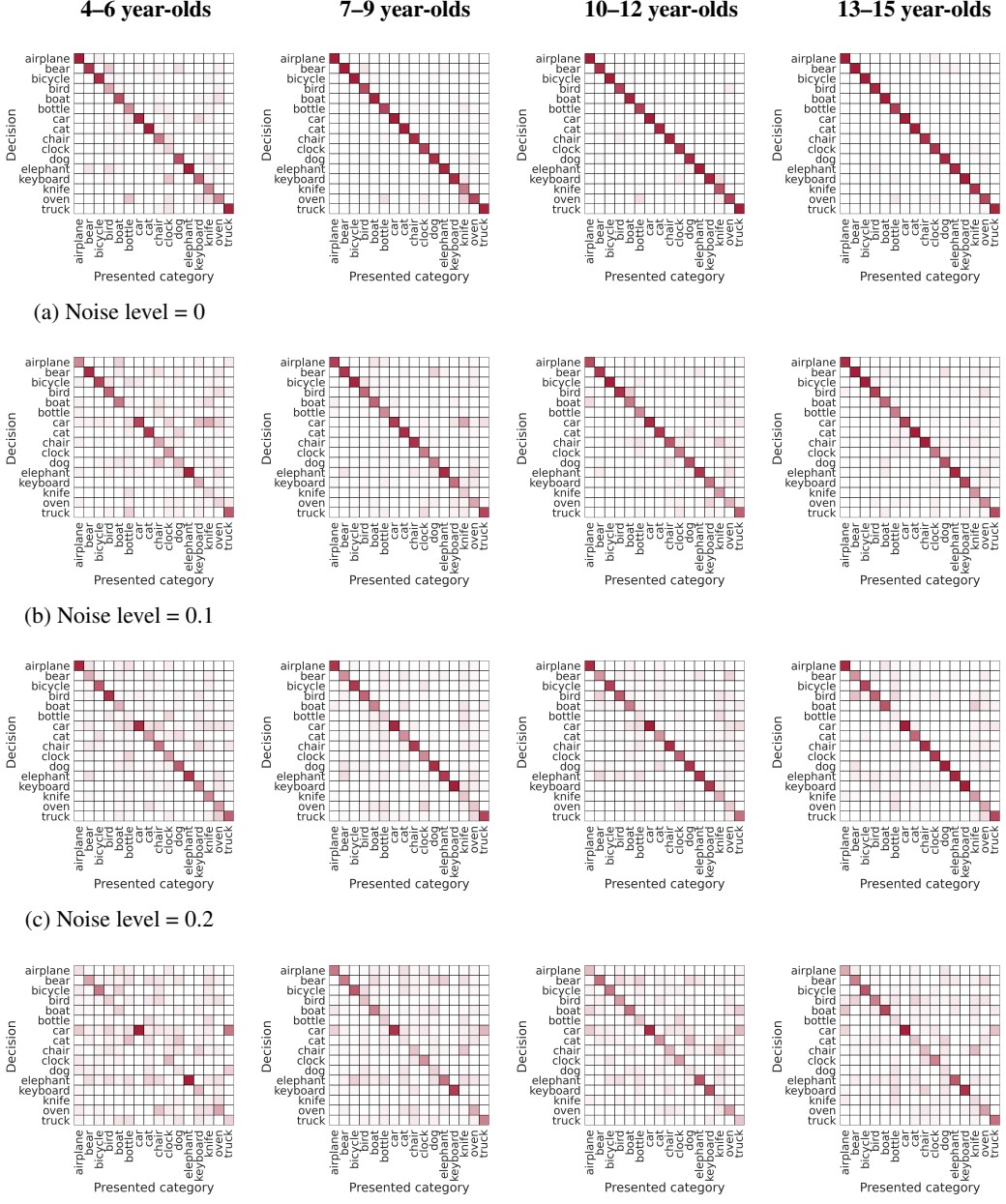

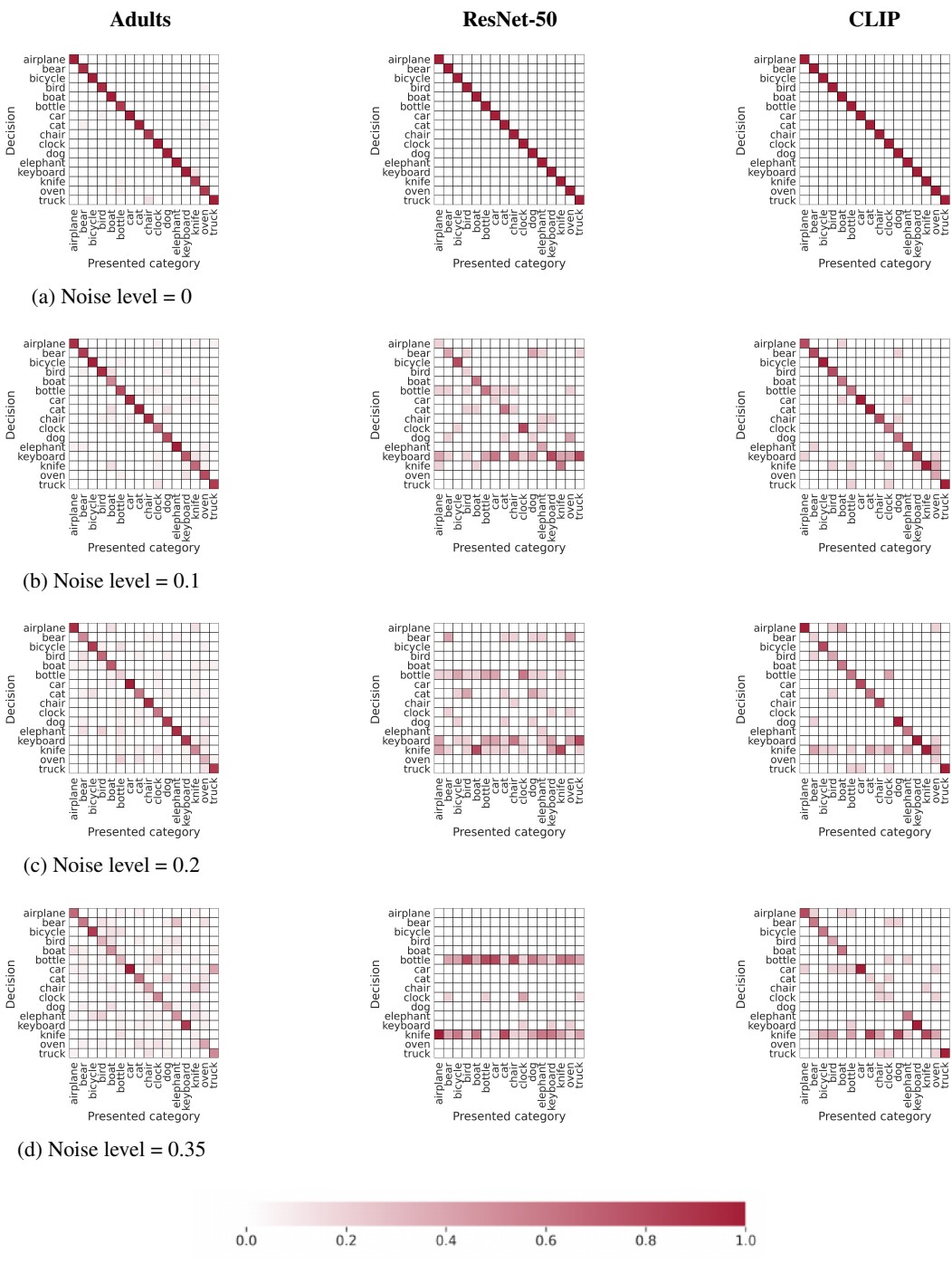

Figure 6: Confusion matrices for all age groups and models across all noise levels (**a-d**). Rows show classification decision of observers and models and columns show the ground truth label of the presented category. Transparency of single squares within a matrix represent response probabilities (fully transparent = 0%, solid red = 100%). Entries along the negative diagonal represent correct responses; entries which are off the negative diagonal indicate errors.

