# OpenReview forum: "A four-year-old can outperform ResNet-50: Out-of-distribution robustness may not require large-scale experience"
_NeurIPS.cc/2021/Workshop/SVRHM — SVRHM 2021 Poster_

### Official Review · Reviewer_9K7V · 2021-10-28

**Rating:** 4
**Confidence:** 4

**Review:**

This paper measures the accuracy with which children can classify images under different levels of noise distortion. They find that, although children’s overall accuracy is lower than that of ML models, their relative accuracy when confronted with salt-and-pepper noise is higher than that of an ImageNet-trained ResNet-50. By comparison, CLIP is only slightly less robust to noise than humans.

I find the paper’s interpretation of these results frustrating. While I believe it’s plausible that there are properties of the human visual system beyond data that make it more robust to noise than our current neural networks, the results presented don’t provide convincing evidence in favor of that hypothesis.

1. The 50M images that children are calculated to have seen is clearly more than the 1.3M images in ImageNet, and multiplying 1.3M by 90 because each image is seen 90 times during training doesn’t seem appropriate. The whole reason to train on large datasets is that training for longer on smaller datasets does not give the same result. Training on ImageNet for 10,000 epochs will clearly not lead to the same robustness as CLIP, which is trained on 300M images for 32 epochs, although both would see about 10 billion images over training. From Table 1 and Figure 4, it’s clear that training ResNet-50 for more than a couple dozen epochs doesn’t yield much improvement in robustness.
2. The retina is intrinsically noisy, and in low light, there is shot noise to contend with as well. The visual system has to learn to be robust to that noise, but a neural network does not. Standard data augmentation for training neural networks crops images but doesn’t generally add noise. However, augmenting images by adding a single type of noise can improve robustness to other types of noise as well; [1] shows that simply training on images corrupted with Gaussian noise can greatly increase robustness under salt-and-pepper noise (called impulse noise there). Furthermore, the number of different noise patterns that the visual system receives is also almost certainly higher than 50M, since retinal integration times are 10-100 ms, not 2 seconds.

The data the authors have collected are unique and I wish I could recommend acceptance, but frankly I think these are big issues that are hard to fix. Point 1 would ideally be addressed by training a very large model on a 50M example dataset, but since that is likely to be difficult, it could potentially be addressed by measuring how robustness varies with the actual training dataset size, fitting a curve to that data, and extrapolating. With point 2, I worry that it will be hard to come up with any conclusive solution, because it is not clear what augmentation would mimic human vision. Using salt-and-pepper noise as augmentation would clearly give the ResNet an unfair advantage, since humans do not actually see a large amount of salt-and-pepper noise, but would at least provide some upper bound on the robustness achievable by changing the augmentation.

Minor comments:

- The behavior of the curves in Figure 2 is a bit weird. The decrease in accuracy from 0.1->0.2 seems similar to the decrease in accuracy from 0->0.1 for the adults, but for the kids, the latter seems larger than the former. It could be noise, but it happens for all three age ranges.
- Related to the above point, Figure 2 should have error bars.
- It is generally possible to get much higher accuracy at early points in training by computing an exponential moving average of the parameters and using this moving average for evaluation, as in [2].

References:

[1] Rusak, E., Schott, L., Zimmermann, R. S., Bitterwolf, J., Bringmann, O., Bethge, M., & Brendel, W. (2020, August). A simple way to make neural networks robust against diverse image corruptions. In European Conference on Computer Vision (pp. 53-69). Springer, Cham.

[2] Szegedy, C., Vanhoucke, V., Ioffe, S., Shlens, J., & Wojna, Z. (2016). Rethinking the inception architecture for computer vision. In Proceedings of the IEEE Conference on Computer Vision and Pattern Recognition (pp. 2818-2826).

---

### Official Review · Reviewer_xnY9 · 2021-10-29
**Interesting experiment though too much is made of estimated quantities**

**Rating:** 6
**Confidence:** 4

**Review:**

This paper compares robustness to a particular corruption type (salt-and-pepper noise) in people (children of different age groups and adults) and models (ResNet-50 and CLIP), and finds that, whereas the performance of ResNet-50 drops off rapidly as a function of corruption level, children of all ages, adults, and CLIP exhibit greater robustness and a similar robustness trajectory. The authors then present (highly speculative) calculations of the amount of visual experience of each group, and compare robustness as a function of number of images seen. The high-level question is interesting and the human experiments were non-trivial to perform, so on balance, it is worth sharing these results in a workshop setting. Nevertheless, there are a number of ways in which the authors could strengthen the work.

**Specific comments**
* This study compares OOD robustness in people and models. However, the stimuli are limited to a single corruption type (salt-and-pepper noise). Given this, I would suggest being more circumspect in conclusions about robustness in general, and noting the limitations as well as which other stimulus types would be useful to investigate in future.
* The experimental methods state that short presentation times were used, but the presentation time appears to be missing.
* To compare coarse-grained classification error patterns, the authors qualitatively compare confusion matrices for the different human groups and models. It would be useful to add a quantitative metric here (e.g. correlation between pairs of confusion matrices as a function of noise level).
* Section 4 is highly speculative, and involves what the authors themselves refer to as "back-of-the-envelope" calculations, for instance estimating the number of images to which children are exposed as a function of age. While the enterprise of comparing robustness in models and people as a function of visual experience is clearly important, and I appreciate that the authors are up front in the fact that they are crudely estimating these quantities, it strikes me that some of the estimates nevertheless require further justification.
    * The human estimates involve two constituent estimates: the number of hours children are awake, and how many images they see per unit time. While the authors provide a citation for the first estimate, the second appears to be purely speculative (the authors ask the reader to "imagine" moving around a room in a footnote). Are there sources of evidence in the literature that the authors can draw from in making this estimate? There are a number of datasets consisting of head-mounted cams collected from children (though possibly of younger age groups than the one in this study), for example Sullivan et al. 2020, Aslin 2012, Bambach 2016, and others. Are any of these datasets useful for these estimates? If not, it would be worth noting the limitations of existing datasets and what would be needed. Second, an observation here is that, for people, visual experience is likely highly correlated over time -- in other words, children may see many views of an object, but not a great many new objects, per unit time. How does this compare to the visual experience of models like ResNet-50?
    * The authors briefly mention (footnote 1) that human learning is largely self-supervised rather than supervised, as in the ResNet-50 they probe: "Thus, in some sense one input image is 'worth' more for a CNN than for a human". If included, this statement needs further elaboration. It also suggests that another comparison, namely with self-supervised models, especially with comparably matched visual experience, might be useful. One data point here is that Zhuang et al. 2021 (PNAS) found that a CNN trained on a self-supervised objective on images from the SAYCam dataset (head-mounted camera on infants, Sullivan et al. 2020) predicted primate neural data as well as a supervised CNN trained on ImageNet. See also Bambach et al. 2017 (IEEE).
    * On the model side, the authors calculate the number of images seen by a ResNet-50 by summing images over training epochs, rather than as the training data size. Related to the above, it would be useful to distinguish here between objects and views (of the same image), and explain which (or both) is of interest in studying robustness, and why. Either way, how were the views generated? I did not see any details about which data augmentation(s) were used in training the ResNet-50, yet model robustness to common corruptions depends on the data augmentations used during training. It will be important to justify the choice of data augmentations in addition to describing them.
* There is relatively little discussion of CLIP -- either of its training data and training objective or of hypotheses about why it better matches people on the metric in this study; this is a lost opportunity.
* In the discussion, the authors claim that "to our knowledge, this is the first study to directly compare children, adolescents, and different neural networks in a core object recognition task". It would be worth adding citations to related existing work which compares object recognition capabilities of models trained on visual experience collected from children versus adults as a function of different features of the training data (e.g. Bambach et al. 2017) (I realize that the present study is different in probing recognition of humans directly in a psychophysics experiment, but this work is highly relevant to especially the questions about what various types of visual experience buy a vision system).
* I found that title confusing: at first, I thought that "limited image exposure of a four-year-old" referred to stimulus presentation time, rather than to amount of visual experience.
* Minor: There are several missing words in the first section.

---

### Official Review · Reviewer_qWuA · 2021-11-01
**Interesting developmental data, questionable comparison of input data for children and CNNs**

**Rating:** 8
**Confidence:** 4

**Review:**

The paper investigates an interesting question and is well written. The presented measurements of the developmental trajectory of human object recognition are original and an important contribution. However, I do have two major concerns regarding the comparisons between humans and models.
1) The estimated number of images seen by different age groups is critical, yet I am not convinced by the estimate in the paper. The authors assume a new image every 2 seconds and state that this a conservative upper bound. I disagree with this notion. Humans make about 3 fixations per second, i.e., 6 images every 2 seconds. Instead of 50 Mio input images for a 4-to-6-year-old human visual system this would mean 300 Mio input images, i.e., much more than the 90*1.3 Mio images during a standard image net training. Even a more conservative estimate of two fixations per second would exceed the number of images during a standard image net training. I agree with the authors that these images might not be “completely different”, because they will often contain similar views of the same objects. This argument seems weak given that ninety epochs of standard image net training also contain 90 identical views of the same images. More importantly, it is possible that the existence of many similar images of the same objects from slightly different perspectives plays an important role in learning a robust visual representation of objects. In my opinion, the conclusion that human robustness does not require large-scale training is therefore false and section 4 of the paper needs to be revised extensively.

2) The authors discuss the two models CLIP (with a ViT-B/32 backbone) and ResNet-50 with regard to their differences in terms of the sheer amount of training images. However, the models differ in more dimensions than the number of training images and the authors should discuss these differences and the implications they have for the interpretation of the study.

Minor comments:
1) The footnote on page 4 refers to differences between humans and CNNs regarding the supervision of the learning process. This is an important topic and should be discussed not just mentioned in a footnote.

2) The two hypotheses stated at the end of the introduction give an interesting theoretical framework, but are neither mutually exclusive nor is the study capable of falsifying either of them.

Typos:
P2. L.47: “to to“

---

### Author Response · Authors · 2021-12-09
**Review response**

Dear anonymous reviewers

Thank you for your valuable comments and inputs. We tried to implement as many of them as possible. As all reviewers remarked quite many similar points, this comment can be seen as a global response to all reviews.

Besides implementing some minor changes throughout the paper, we completely revised Section 4:

We tried to make our estimates as transparent as possible.

Based on empirical data we calculate fixations per second for all age groups individually.

We do no longer compare to the total number of images a model has encountered (dataset size times epochs) but only to dataset size.

We scale down the total number of fixations humans have made during their lifetime by a certain factor to get the number of input images and allow for fair comparison.

By scaling down by two different factors, we provide a lower and an upper bound for those estimates.

For more exhaustive comparison, we evaluated  seven more models on our OOD dataset, including a current self-supervised model.

We elaborated on toddler inspired real world data as input for deep learning models.

And we elaborated on the role of object view diversity vs. general dataset size.

---

### Decision · Program_Chairs · 2021-11-02

Accept (Poster)